# Prevalence of Chronic Heart Failure, Associated Factors, and Therapeutic Management in Primary Care Patients in Spain, IBERICAN Study

**DOI:** 10.3390/jcm10184036

**Published:** 2021-09-07

**Authors:** Jose L. Llisterri-Caro, Sergio Cinza-Sanjurjo, Vicente Martín-Sánchez, Gustavo C. Rodríguez-Roca, Rafael M. Micó-Pérez, Antonio Segura-Fragoso, Sonsoles Velilla-Zancada, Jose Polo-García, Alfonso Barquilla-García, Luis Rodríguez Padial, Miguel A. Prieto-Díaz

**Affiliations:** 1Spanish Society of Primary Care Physicians (SEMERGEN)’s Foundation, 28009 Madrid, Spain; jllisterric@gmail.com; 2Porto do Son Health Center, Health Area of Santiago de Compostela, 15970 Santiago de Compostela, Spain; 3Institute of Biomedicine (IBIOMED), University of León, 24004 León, Spain; vicente.martin@unileon.es; 4Epidemiology and Public Health Networking Biomedical Research Center (CIBERESP), 24004 León, Spain; 5La Puebla de Montalbán Health Table Center, 45516 Toledo, Spain; gcrodriguezroca@gmail.com; 6Fontanars dels Alforins Health Center, Xàtiva–Ontinyent Department of Health, 46635 Valencia, Spain; rafaelmmicoperez@gmail.com; 7Medicine Department, University of Castilla-La Mancha, 13003 Toledo, Spain; asegurafr@gmail.com; 8Joaquin Elizalde Health Center, 26004 Logroño, Spain; svelizan@hotmail.com; 9Casar de Cáceres Health Center, 10190 Cáceres, Spain; jpolog@telefonica.net; 10Trujillo Health Center, 10200 Cáceres, Spain; alfonso.barquilla@gmail.com; 11Servicio de Cardiología, Complejo Hospitalario de Toledo, 45071 Toledo, Spain; lrpadial@gmail.com; 12Vallobín-La Florida Health Center, 33012 Oviedo, Spain; maprietodiaz@telefonica.net

**Keywords:** primary care, chronic heart failure, prevalence, drug treatment

## Abstract

Background: The prevalence of chronic heart failure (CHF) in patients assisted in primary care is not well known. We investigated the prevalence of CHF, its associated factors, and its therapeutic management. Methods and findings: This was a cross-sectional, multicenter study conducted in primary care (PC) in baseline patients of the IBERICAN study (Identification of the Spanish Population at Cardiovascular and Renal Risk). CHF was defined as the presence of this condition in the medical history, classifying patients according to the type of ventricular dysfunction in CHF with preserved ejection fraction (pEF), or CHF with reduced ejection fraction (rEF). Clinical characteristics, relationship between CHF and main cardiovascular risk factors (CVRF), and drug treatments used according to ejection fraction (EF) were analyzed. Results: A total of 8066 patients were included (54.5% women), average age (SD) was 57.9 (14.8) years, of which 3.1% (95% CI: 2.3–3.7) presented CHF, without differences between men and women. CHF with pEF (61.8%; 95% CI: 55.5–67.6) was more frequent in women, and CHF with rEF (38.1%; 95% CI: 33.2–45.5) (*p* = 0.028) was similar in both genders (65.9%; 95% CI: 57.1–73.4 vs. 57.3%; 95% CI: 47.7–65.8) (*p* = 0.188). A progressive increase of the prevalence with age (15.2% in ≥80 years) and with the aggregation of CVRF was observed. The most prescribed treatments were beta-blockers (54.7%) followed by angiotensin converting enzyme inhibitors (42.8%) and angiotensin II receptor antagonists (41.3%), without differences between pEF and rEF. The variables that are most associated with the probability of suffering CHF were a personal history of left ventricular hypertrophy (OR: 5.968; *p* < 0.001), of atrial fibrillation (OR: 3.494; *p* < 0.001), and of peripheral vascular disease (OR: 2.029; *p* < 0.001). Conclusions: Three in every 100 patients included in the IBERICAN study presented CHF, of which two thirds had pEF. The condition increased exponentially with age and aggregation of CVRF. We did not find any differences in drug treatment according to the type of ventricular dysfunction. The treatment of HF with rEF has much room for improvement.

## 1. Introduction

Chronic heart failure (CHF) is a highly prevalent condition and has a great clinical significance. Its poor prognosis, with high mortality rates—in some cases even higher than the death rates of some malignant tumors—and its morbidity, with numerous and costly hospitalizations, together with its progressively increasing incidence, make this disease one of the most relevant social and health problems at the present time [1]. In Spain, the results of the PRICE (Prevalence of Heart Failure in Spain) [2] and EPISERVE (heart failure in outpatient consultations: comorbidities and diagnostic-therapeutic performance by different specialists) [3] population-based studies indicate that approximately 5% of adults over the age of 45 suffer CHF, reaching 16.1% among persons over the age of 75. Additionally, the registers on CHF carried out in Spain offer relevant information on the clinical characteristics of these patients in our setting. In primary care (PC) [4,5], patients with preserved ejection fraction (pEF) are more frequent, more associated with older women with a history of arterial hypertension (AH), whereas in cardiology departments [6], it is more common in middle-aged men with reduced EF (rEF) caused by ischemic heart disease (IHD).

These registers are important sources of information in order to assess the compliance of drug treatments with the recommendations of clinical practice guidelines (CPG) [7]. And this should be so because the treatment of CHF with rEF has undergone major changes in recent years, which has allowed for the optimization of the treatment to improve the prognosis—especially regarding hospitalizations and mortality associated with this condition. Numerous studies and publications demonstrate the benefit of using drugs such as angiotensin-converting enzyme inhibitors (ACEI) [8], beta-blockers (BB) [9], and mineralocorticoid receptor antagonists (MRA) in the treatment of CHF, but in fact these drugs are underused in everyday clinical practice. Studies conducted in PC in Spain indicate this, showing that compliance with CPG is insufficient and differs depending on the healthcare geographical area, accessibility to complementary tests, and training of professionals in CHF [10,11].

Thus, from the published data, we can say that CHF with rEF is a condition whose treatment can be improved, especially in PC, which represents a missed opportunity to prevent complications associated with it. Regular follow-up, review of the treatment, and, if applicable, compliance with the recommendations of CPG are crucial in the disease’s prognosis.

It therefore seems necessary to obtain updated information, in real life conditions of clinical practice, on the prevalence of CHF in PC, as well as to determine the most frequent type of ventricular dysfunction treated by family doctors and the drug treatments received by patients. It is also necessary to assess the influence of different clinical variables, especially the impact of the aggregation of the main cardiovascular risk factors (CVRF) on the risk of developing CHF.

The IBERICAN study (Identification of the Spanish Population at Cardiovascular and Renal Risk) is an extensive study, currently in the follow-up phase, whose primary objective is to find out the prevalence, incidence, and geographical distribution of CVRF in Spanish adult population seen in Primary Care [12]. One of the secondary objectives, as the purpose of this study, was to know the prevalence of CHF, and to determine its associated factors and its therapeutic management.

## 2. Materials and Methods

### 2.1. Design of the Study

The IBERICAN study is an epidemiological, multicenter, observational, prospective study, which is being conducted in primary care in Spain in patients of the National Health System. Using non-probability consecutive sampling, individuals of either sex were recruited, aged between 18 and 85, seen in primary care centers in Spain, who were users of the Spanish National Health System, and who had stable residence in Spain during the last 5 years. Patients with plans to change their place of residence within the following 6 months, terminally ill or with reduced life expectancy for the next 5 years, or refusing to participate were excluded. The design details and characterization of the population have already been published [12].

The field work for this sub-study was carried out between June 2014 and December 2018. A total of 519 family doctors participated, who selected at least 10 patients each by consecutive sampling.

All the patients signed the corresponding informed consent form before their inclusion in the project. The study was classified by the Spanish Medicines and Health Products Agency (AEMPS) as an Observational Not Post-Authorization Study (Not PAS) on 23 January 2013. It was approved by the Clinical Research Ethics Committee (CREC) of the Hospital Clínico San Carlos in Madrid on 21 February 2013 (C.P. IBERICAN-C.I.13/047-E) and it is registered at https://clinicaltrials.gov with the number NCT02261441 (last access: 25 August 2021). The results provided in this paper correspond to the cross-sectional analysis of the patients who had had the inclusion visit by 15 December 2018.

### 2.2. Data on Patients

Socio-demographic variables, CVRF, target organ damage (TOD), and associated cardiovascular disease (CVD) were recorded, in accordance with the guidelines of the European Societies of Hypertension and Cardiology (ESH/ESC) of 2013 [13]. A total of ten classic CVRF were evaluated: age, AH, obesity, abdominal obesity, active smoking, diabetes mellitus (DM), increased LDL-cholesterol, low HDL-cholesterol, hypertriglyceridemia, and sedentary lifestyle. Age >65 years in men and >55 years in women were considered as risk ages. The definition of the study variables has already been published in previous papers [12]. The serum biochemistry was done in the local laboratory at the moment of inclusion. Previous lab tests were considered valid if performed within the last six months.

### 2.3. Data on the Evaluation of Heart Failure

CHF was defined as the inclusion in the medical history of that condition, specifying the classification of patients according to the type of ventricular dysfunction in CHF with pEF (LVEF ≥ 50%) or CHF with rEF (LVEF < 50%), according to the information about the echocardiogram included in the medical record [12].

Left ventricular hypertrophy, as a precursor lesion for CHF, was established on the basis of the information provided by the investigator, who could use electrocardiogram diagnosis (Sokolow–Lyon index > 3.5 mV; RaVL > 1.1 mV; Cornell voltage product >244 mV × ms), and/or echocardiogram diagnosis (>15 g/m, in men; >95 g/m, in women).

### 2.4. Data on Drug Treatment

The class and number of therapeutic subgroups of drugs used by patients were recorded: ACEI, angiotensin II receptor blockers (ARB), loop diuretics, thiazide diuretics, mineralocorticoid receptor antagonists (MRA), BB, statins, antidiabetics, antiplatelet agents, and anticoagulants. The adequacy to drugs recommended by the CPG for CHF with rEF was assessed with two indicators, a general indicator of use of ACEIs or ARBs plus BB plus MRA, or partial when the patient received treatment with two of the previous drugs.

### 2.5. Control of the Main Cardiovascular Risk Factors

Control of AH was considered good (optimal control) in all subjects, including diabetic patients, when systolic blood pressure (SBP) and diastolic blood pressure (DBP) were lower than 140 and 90 mmHg, respectively. AH control was considered excellent when SBP was between 120–130 mmHg and DBP was between 70–80 mmHg, also for DM [14]. Hypotension was considered when SBP was <100 or DBP was <60 mmHg. DM optimal control was considered as HbA1c < 7% [13].

### 2.6. Quality of Data

The process of data collection from the case report forms (CRF) was carried out using remote capture methodology: e-clinical. This process perfects data entry techniques by means of filters in the variables of the CRF, thus reducing errors and reducing time between the data collection processes and the publication of study findings. The investigator accessed a public Uniform Resource Locator (URL) from the Internet, which required him to identify as a member of the research community of SEMERGEN (Spanish Society of Primary Care Physicians) with a username and password. Once validated, he accessed a data collection system implemented under secure servers where the information of patients was encrypted and audited.

### 2.7. Statistical Analysis

All the analyses were performed from one sample of valuable patients which included all those who met the selection criteria. The analyses related to the main variable considered the valuable patients who presented information in the main variable. Results were expressed as frequencies and percentages for qualitative variables, and as means (standard deviation) for quantitative variables. The 95% confidence interval (95% CI) was calculated for the variables of interest, assuming normal distribution and using the exact method for small proportions [15].

For the comparison of subgroups of patients, parametric tests (Student’s *t*-test or ANOVA) or non-parametric tests (Mann–Whitney or Kruskal–Wallis) were used for quantitative variables, according to the specific characteristics of the variables examined. The Chi-square test was used for qualitative variables. The unconditional logistic regression method stepwise backward was used to determine which variables were associated with the presence of CHF. In order to evaluate which factors were associated with CHF independently, a binary logistic regression model was built, with a bilateral significance level of 0.05 for all statistical tests. The multivariate model was built by initially including all the variables (33 variables) which showed bivariate association with CHF (*p* < 0.05). From this model, non-significant variables were removed manually until a final model with 11 variables independently associated with CHF was achieved. Adjusted odds ratios (OR) and their 95% confidence interval were presented. In all contrasts, the null hypothesis was rejected when the alpha error was lower than 0.05. The statistical package SPSS 23.0 (Statistical Package for Social Sciences) for Windows (Armonk, NY, USA: IBM Corp. Released 2013. IBM SPSS Statistics for Windows, version 23.0 Armonk, NY: IBM Corp) was used for the data analysis.

## 3. Results

### 3.1. Description of the Sample

A total of 8066 patients were included (54.5% women), with an average age (SD) of 57.9 (14.8) years. The most frequent CVRF were abdominal obesity (55.6%), dyslipidemia (50.3%), and AH (48.0%); 20.2% of patients had DM. The most frequent subclinical TOD was microalbuminuria (7.6%) and a glomerular filtration rate (GFR) < 60 mL/min (7.3%). A total of 16.3% of the sample presented prior CVD, IHD being the most frequent (44.5%). The complete social, health, and clinical characteristics of the sample have been previously published [12].

### 3.2. Prevalence of Chronic Heart Failure

In total, 3.1% (95% CI: 2.3–3.7) of patients included presented CHF; 3.2% (95% CI: 2.6–3.8) in men and 3.0% (95% CI: 2.5–3.5) in women (*p* = 0.61). The prevalence of CHF with pEF was significantly higher than CHF with rEF (61.8%; 95% CI: 55.5–67.6 vs. 38.1%; 95% CI: 33.2–45.5) (*p* = 0.028). CHF with pEF was more frequent in women than in men (65.9%; 95% CI: 57.1–73.4 vs. 57.3%; 95% CI: 47.7–65.8) (*p* = 0.188), and CHF with rEF was more frequent in men (43.6%; 95% CI: 34.4–52.6 vs. 35.6%; 95% CI: 27.4–44.0) (*p* = 0.161).

The prevalence of CHF increased progressively with age, both in men and in women, from 1.2% in patients <65 years of age, 4.7% between 65–80 years, up to 15.2% in patients older than 80 (*p* < 0.001) (Table 1). In patients without CVD, the prevalence increased as CVRF were associated (Figure 1), from 0% in patients with no CVRF, up to 12.4% in patients with 8–10 CVRF (*p* < 0.001), Appendix A.

### 3.3. Control of Main Cardiovascular Risk Factors

The average values of SBP and DBP were, respectively, 134.4 and 76.3 mmHg. We observed an optimal control of BP in 51.5% of patients, and an excellent control in 26.5%. A total of 43.7% of patients showed an optimal control of HbA1C, Appendix A.

### 3.4. Drug Treatment for CHF

The most prescribed treatments according to EF were BB (54.7%), followed by ACEI (42.8%), ARB (41.3%), loop diuretics (34.1%), and thiazide diuretics (27.0%), without significant differences between pEF and rEF. In CHF with rEF, the simultaneous use of ACEI (or ARB) + BB + MRA was made by 9.1% of patients; the combination of ACEI (or ARB) + BB by 45.5%; ACEI (or ARB) + MRA by 14.3%; and BB + MRA by 14.3%, Appendix A.

### 3.5. Variables Associated with Chronic Heart Failure

According to the binary logistic regression model, the variables which are likely to associate independently with CHF were age (<64 years, 65–79 years, 80 or older), male/female gender, CVRF (smoking, sedentary lifestyle, obesity, abdominal obesity, DM, AH, dyslipidemia), family history of CVD, TOD, associated clinical disease (CVD, IHD, peripheral arterial disease, advanced retinopathy), analytical parameters, drug treatments (AH, DM, dyslipidemia), time from onset of AH, time from onset of DM, SPB control, DBP control, and DM control. Table 2 shows the variables resulting from the final model. The strongest independent association was observed with left ventricular hypertrophy (OR: 5.968; *p* < 0.001), and with a personal history of atrial fibrillation (OR: 3.494; *p* < 0.001) and of PVD (OR: 2.029; *p* < 0.001). A negative relationship was found with the progressive increase of GFR (OR: 0.988; *p* < 0.05).

## 4. Discussion

The results of the IBERICAN study, conducted in a large population of patients visiting the first level of care of the Spanish Health System, show that the prevalence of CHF reaches 3%, without differences between men and women. This condition increased exponentially with age and with the aggregation of CVRF.

The study includes a homogeneous sample, with sociodemographic and clinical characteristics very similar to other studies [16,17,18], which presumably reflects patients over 18 years of age of PC Health Centers. It shows a slight predominance of women, who frequently present a history of AH, obesity, and other CVRF. This clinical profile differs from hospital patients, where younger and primarily male patients predominate [3,6,19].

To our knowledge, no study to date has evaluated the prevalence of CHF, its association with the aggregation of CVRF, and its therapeutic management in a large population sample treated exclusively in PC.

The prevalence of CHF found in the scientific literature varies according to the healthcare setting, to the age of the population included, and to the methodology used. For instance, in the population-based study PRICE [2], conducted in PC and cardiology, the prevalence was 6.8%, similar in men and women. In the EPISERVE [3] study, carried out in patients of PC (*n* = 778), cardiology (*n* = 777), and internal medicine (*n* = 694), the prevalence of CHF in the total of patients was 4.7% (2% in PC, 17% in cardiology, and 12% in internal medicine). Some studies produced similar data [20], and other studies based on computer records described significantly lower prevalence figures, about 1% [21,22]. These values obtained from computer records probably underestimate the prevalence because they depend on the correct coding of diagnoses, as well as the well-known inter-institutional variability and the unreliability of CHF diagnoses in administrative records [23]. Studies conducted in PC from outside Spain show CHF prevalence ranging from 1.2% in the Netherlands [24] to 4.3% in Portugal [25].

In the IBERICAN study, as has been repeatedly described [26], CHF increased with age, in line with what has been confirmed in PC Spanish population [4,5,18] and in European studies [24,25,27]. In our study, the prevalence reached 15.2% of patients over the age of 80. In the PRICE study [2], by age, the prevalence was 1.3% in patients aged 45–54, 5.5% in patients aged 55–64, 8% in patients aged 65–75, and 16.1% in patients over 75, similar to our percentages and to those found in European studies [24,25,27].

As could be expected, CHF increased its prevalence as patients’ cardiovascular risk increased and with the coexistence of the main CVRF [28,29]. In our study, the CVRF more frequent in patients with CHF were AH, dyslipidemia, abdominal obesity, and DM. In addition, as CVRF were aggregated in subjects without CVD, on the contrary, the patients who did not have CVRF did not present CHF. These findings make it clear that the aggregation of CVRF is crucial in the evolution of the disease.

The average values of SBP and DBP were, respectively, 134.4 and 76.3 mmHg, and were controlled in 48.5% of patients, this percentage being identical to the one found in the GALICAP [4] study and very similar to other studies carried out in PC [5,30]. Although the design of this study does not allow for conclusive relationships regarding determinants of poor control, it is very likely to be linked with the clinical characteristics of the subjects included, who were at an advanced age, predominantly women, and with a high prevalence of AH, DM, and associated diseases, factors known as generators of poor control [13].

The prevalence of CHF with pEF was significantly higher. These results are in line with other studies conducted in PC [4,18,28], in which, as in our study, there is a predominance of older patients with a higher proportion of women, who often present a history of AH, abdominal obesity, and other CVRF, and, as a result of AH, present more TOD, such as left ventricular hypertrophy and higher pulse pressure.

Patients with CHF and rEF of the IBERICAN study, as described in hospital-based [6] and PC-based [4,5,18] studies, were younger and mostly men, and mainly had a history of IHD. They also had more PVD and more renal involvement (GFR < 30 mL/min and microalbuminuria), factors connected with CHF for decades [31,32].

The analysis of drug treatments shows that patients with CHF received a more intense treatment for the main CVRF (AH and dyslipidemia) and more antiplatelets and anticoagulants. This is because the population with CHF presented a higher prevalence of CVRF, TOD, and associated CVD, which may dictate a greater therapeutic intensity. In patients with CHF, the most prescribed treatments were renin-angiotensin system blockers, followed by BB, loop diuretics, and thiazide diuretics, without significant differences between treatments applied depending on the type of CHF. Our results showed that the adherence to the GPC was poor because less than 10% of the patients with CHF and rEF are taking the recommended treatment. This finding, found also in other studies [3,4,11,33,34], indicates that the prescription of drugs for CHF with rEF is not optimal and, therefore, that the recommendations of the CPG for the use of renin-angiotensin-aldosterone system blockers and of BB have not yet been implemented in Spanish PC. This differs from patients with CHF treated in hospital cardiology departments in Spain, where compliance with the guidelines has proved to be excellent [35]. In our analysis, it stands out that the treatment in patients with pEF is similar to patients with rEF, even though the CPG do not indicate any particular treatment, given that none has been shown to reduce mortality in this clinical scenario [7]. This might be explained by the fact that the only recommendation is a good control of CVRF; for this, both ACEIs and ARBs are useful, and therefore it is possible that the physician preferably uses this type of drugs in this scenario, drugs which are widely established in our country. Regarding BBs, their use could be determined by ischemic heart disease or atrial fibrillation, which, although more frequent in patients with rEF, the total number of cases increases in both groups of CHF. This could consequently increase the number of patients who are treated with this therapeutic group and could be the reason why there are no statistically significant differences.

In the multivariate analysis, after adjustment for the rest of variables, CHF was associated with older age, presence of left ventricular hypertrophy, and a history of atrial fibrillation and of PVD. The association with these variables is well known [35] and is consistent with findings in other studies [5,18,28,36].

Some limitations should be taken into account when assessing this work. First, the selection of doctors and patients has not been random, which does not enable to strictly generalize our results to the population seeking care in PC. Second, the diagnosis of CHF, specifying the type of dysfunction, was made using records in the medical history, without checking against echocardiographic or other reports. Likewise, the biochemical tests were not analyzed centrally, but in the reference laboratory of each investigator, so some small differences in the assessment cannot be excluded due to a different calibration of the analyzers. Third, with the current increased cancer survival rates, chemotherapy has become a rising cause of ventricular dysfunction; this variable is not recorded in our study, and therefore we cannot quantify the impact that it may have on our sample.

Unfortunately, we do not have data on prescription of new treatments which have been shown to reduce cardiovascular morbidity and mortality in patients with CHF and rEF, such as the combination sacubitril/valsartan or sodium-glucose cotransporter 2 (SGLT2) inhibitors.

Nor do we have data which breaks down the use of spironolactone or eplerenone because the data record was made on the basis of potassium-sparing diuretics. The beginning of the major inclusion of patients in 2014 (the PARADIGM study, the DAPA-HF trial and the EMPEROR-reduced trial were published late in this year [37], and the CPG of the ESC in 2016 [7]) and the delay in the implementation of therapeutic recommendations of the CPG have caused the absence of record of these variables.

However, the size of the population analyzed makes the IBERICAN study the most comprehensive of the studies conducted in PC in Spain. The data collected, with few rejections/exclusions, leads us to believe that the results have good internal validity, providing a reasonable approximation to the real situation of patients treated in everyday PC clinical practice.

## 5. Conclusions

In conclusion, the results of the IBERICAN study indicate that 3% of patients seen in PC in Spain present CHF. This disease is more prevalent in older patients and increases exponentially with the aggregation of CVRF. Left ventricular hypertrophy, receiving anticoagulant treatment, as well as a history of atrial fibrillation and of PVD are the variables most associated with the condition. The findings of the IBERICAN study suggest, in line with previous research, that we are facing a disease related to aging and to the aggregation of CVRF. CHF with pEF is more frequent than CHF with rEF, noting that, unfortunately, there has been no improvement in the drug treatment of this type of ventricular dysfunction in PC in recent years. It is necessary to promote training and implementation of the CPG in order to modify the current therapeutic situation of CHF, especially of CHF with rEF. The cohort of the IBERICAN study, presently in a five-year follow-up phase, will probably contribute to give a satisfactory response on these issues.

## 6. Patents

Any patents resulting from the work were reported in this manuscript.

## Figures and Tables

**Figure 1 jcm-10-04036-f001:**
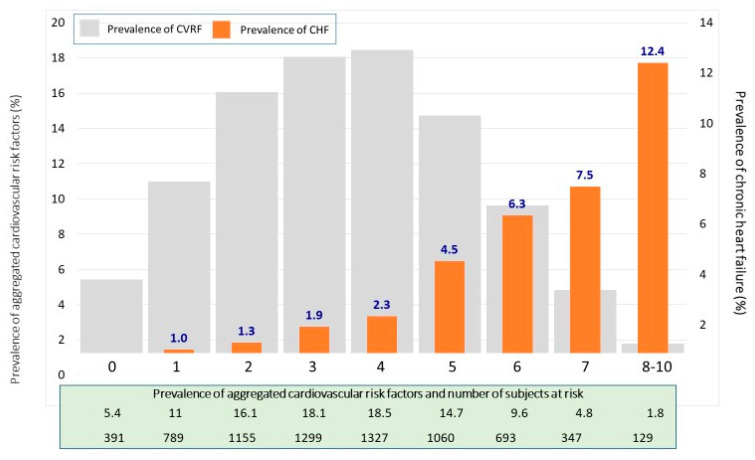
Prevalence of chronic heart failure according to aggregation of cardiovascular risk factors in patients without established cardiovascular disease. Evaluated cardiovascular risk factors: age (>65 years in men and >55 years in women), diabetes mellitus, arterial hypertension, general obesity, abdominal obesity, active smoking, increased LDL-cholesterol, low HDL-cholesterol, hypertriglyceridemia, and sedentary lifestyle. CVRF: cardiovascular risk factors; CHF: chronic heart failure.

**Table 1 jcm-10-04036-t001:** Prevalence of chronic heart failure by groups of age and gender.

	*n*	Prevalence (%)	95% CI	*p*
All 18–50 years 51–64 years 65–79 years ≥80 years	249	3.1	2.31–3.76	
	20	0.9	0.52–1.31	<0.001
	43	1.6	1.14–2.11	
	120	4.7	3.93–5.62	
	66	15.2	11.98–18.93	
Men (all) 18–50 years 51–64 years 65–79 years ≥80 years	117	3.2	2.64–3.81	
	9	0.9	0.42–1.74	<0.001
	20	1.6	0.95–2.39	
	60	5.0	3.86–6.42	
	28	13.1	8.91–18.35	
Women (all) 18–50 years 51–64 years 65–79 years ≥80 years	132	3.0	2.51–3.54	
	11	0.8	0.40–1.42	<0.001
	23	1.6	1.00–2.36	
	60	4.5	3.41–5.68	
	38	17.3	12.52–22.82	

**Table 2 jcm-10-04036-t002:** Variables associated to chronic heart failure in total population *.

Variables	Odds Ratio	95% CI	*p* ^1^
GFR_CKD_EPI (continuous per unit)	0.988	0.978–0.998	0.014
Triglycerides (continuous per unit)	1.003	1.001–1.004	0.005
BMI (continuous per unit)	1.028	1.007–1.050	0.009
Age (continuous per year)	1.034	1.012–1.057	0.002
Arterial hypertension (yes vs. no)	1.421	1.196–1.688	<0.001
Personal history of PVD (yes vs. no)	2.029	1.206–3.414	<0.001
Personal history of AF (yes vs. no)	3.494	2.018–6.049	<0.001
TOD_LVH (yes vs. no)	5.968	3.960–8.994	<0.001

* Multivariate logistic regression, stepwise backward method (LR). CI: confidence interval; *p* ^1^: statistical significance, Wald’s Chi-squared test; GFR: estimated glomerular filtration rate by CKD-EPI; BMI: body mass index; PVD: peripheral vascular disease; AF: atrial fibrillation; TOD: target organ damage; LVH: left ventricular hypertrophy.

## Data Availability

The data presented in this study are available on request from the corresponding author.

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
