# Peer review of "Prevalence of Chronic Heart Failure, Associated Factors, and Therapeutic Management in Primary Care Patients in Spain, IBERICAN Study"

_jcm, 2021, doi:10.3390/jcm10184036_

Round 1
Reviewer 1 Report
The authors present an interesting study on the prevalence of heart failure in primary care patients. The prevalence of main cardiovascular risk factors was also analyzed and their association with heart failure was investigated. Some comments:
1. The authors repeat the results in the discussion. This is particularly evident in the case of pharmacological treatment, but also when discussing multivariate analysis. Repetitions should be removed from the discussion.
2. The authors discuss in detail the use of drugs depending on the type of heart failure, while admitting that the data collected on pharmacological treatment is incomplete due to the method of data collection and the time of conducting the study. They admit also, that the diagnosis of heart failure was based on the analysis of medical data and was not verified. Therefore, a detailed discussion of the treatment used in different types of heart failure does not seem to be appropriate.
3. In the part of the discussion on blood pressure, they themselves admit that due to the design of the study, the cause of insufficient blood pressure control cannot be reliably established. In the next sentence, however, they speculate that the reasons may be similar to those cited in the available literature.
In the opinion of the reviewer, this part of the discussion should be removed.
Author Response
The manuscript had a mistake, since the data in Table referenced as 3 is actually included in the supplementary material as Tables S1 and S2. We attach a new version of the manuscript.
We sent on 17th august a new version by mail. we attached it now by the website.

Reviewer 2 Report
The study investigates the prevalence of congestive heart failure (CHF) and its associated factors including over 8,000 patients enrolled in the IBERICAN study.
The authors found a CHF prevalence of 3% (2/3 CHF with pEF) and increased risk of CHF in elderly patients and in patients with more cardiovascular risk factors. These findings are interesting and the paper is well-written. Although some of these findings are not novel, it remains important to re-evaluate prognosis of CHF patients due to continuous improvement of cardiovascular therapies (pharmacological treatment, Revascularization strategies and ICD) leading to significant changes of characteristics of CHF patients during past decades.
Comments:
- Baseline characteristics of the study population should be demonstrated comparing patients with and without CHF as well as those with rEF compared to pEF.
- Over 80% were treated with ACEi or ARB and one 51% with beta blockers. The authors should comment on this discrepancy compared to other studies in this field, suggesting much better guideline adherence.
- Do the authors have information on outcomes of CHF patients. What were predictors of mortality? Kindly add and comment on this.
- Do the authors have information on procedural data (coronary angiogram, device therapies)?
- What were etiologies of CHF? Kindly provide further insights.
Author Response
Over 80% were treated with ACEi or ARB and one 51% with beta blockers. The authors should comment on this discrepancy compared to other studies in this field, suggesting much better guideline adherence.
In the discussion of the manuscript, we comment our results regarding the use of drugs and other studies in our country, even observing differences between results in Primary Care and in Cardiology Services: This finding, found also in other studies [3,4,11,33,34], indicates that the prescription of drugs for CHF with rEF is not optimal and, therefore, that the recommendations of the CPG for the use of renin-angiotensin-aldosterone system blockers and of BB have not been yet implemented in Spanish PC. This differs from patients with CHF treated in hospital cardiology departments in Spain, where compliance with the guidelines has proved to be excellent [35].
In the same way, we make a comment regarding the similarity in the drug treatment in patients with pEF and rEF: In our analysis, it stands out that the treatment in patients with pEF is similar to patients with rEF, even though the CPG do not indicate any particular treatment, given that none has been shown to reduce mortality in this clinical scenario [7].
We think that both comments would be sufficient to explain and compare our results.
Do the authors have information on outcomes of CHF patients. What were predictors of mortality? Kindly add and comment on this.
We currently have follow-up data with a median of approximately 30 months. We would like to have more follow-up data to be able to obtain reliable information on follow-up of patients with HF and at the same time of rHF and pHF. However, it may also be interesting to have data in real life depending on the degree of adherence to the therapeutic regimens recommended by the Clinical Practice Guidelines. In any case, from the Steering Committee we consider that the data will be reliable and consistent when we have more time to follow up of our cohort.
Do the authors have information on procedural data (coronary angiogram, device therapies)?
What were etiologies of CHF? Kindly provide further insights.
We don´t have information about procedural data because in the protocol of the study we didn´t recording this information.
About the aetiologies we analysed in both supplementary tables the comorbidities more frequent in both groups. And, after revision of these tables, arterial hypertension and diabetes mellitus are the causes more frequent in patients con HF, but in patients with rHF only the ischaemic heart disease is more prevalent. We think that both tables are enough to show the causes, but if the reviewer consider necessary a new table we can design it to explain the causes.

Round 2
Reviewer 2 Report
Thank you for revising the manuscript as suggested. I do not have further questions regarding your submission, congrats!